# Increased mobilization of mesenchymal stem cells in patients with acute respiratory distress syndrome undergoing extracorporeal membrane oxygenation

Christian Patry[1]☯, Thalia Doniga[2]☯, Franziska Lenz[3], Tim Viergutz[3], Christel Weiss[4], Burkhard Tönshoff[1], Armin Kalenka[3,5], Benito Yard[6], Jörg Krebs[3], Thomas Schaible[2], Grietje Beck[7], Neysan Rafat[1,2,8]*

1 Department of Pediatrics I, University Children's Hospital Heidelberg, University of Heidelberg, Heidelberg, Germany, 2 Department of Neonatology, University Children's Hospital Mannheim, University of Heidelberg, Heidelberg, Germany, 3 Department of Anaesthesiology and Intensive Care Medicine, University Medical Center Mannheim, University of Heidelberg, Heidelberg, Germany, 4 Department of Medical Statistics and Biomathematics, Medical Faculty Mannheim, University of Heidelberg, Heidelberg, Germany, 5 Department of Anaesthesiology and Intensive Care Medicine, Hospital Bergstraße, Heppenheim, Germany, 6 Department of Medicine V, University Medical Center Mannheim, University of Heidelberg, Heidelberg, Germany, 7 Department of Anaesthesiology and Intensive Care Medicine, Dr. Horst-Schmidt Clinic, Wiesbaden, Germany, 8 Department of Pharmaceutical Sciences, Bahá'í Institute of Higher Education (BIHE), Teheran, Iran

☯ These authors contributed equally to this work.
* Neysan.Rafat@umm.de

## Abstract

### Background

The acute respiratory distress syndrome (ARDS) is characterized by pulmonary epithelial and endothelial barrier dysfunction and injury. In severe forms of ARDS, extracorporeal membrane oxygenation (ECMO) is often the last option for life support. Endothelial progenitor (EPC) and mesenchymal stem cells (MSC) can regenerate damaged endothelium and thereby improve pulmonary endothelial dysfunction. However, we still lack sufficient knowledge about how ECMO might affect EPC- and MSC-mediated regenerative pathways in ARDS. Therefore, we investigated if ECMO impacts EPC and MSC numbers in ARDS patients.

### Methods

Peripheral blood mononuclear cells from ARDS patients undergoing ECMO (n = 16) and without ECMO support (n = 12) and from healthy volunteers (n = 16) were isolated. The number and presence of circulating EPC and MSC was detected by flow cytometry. Serum concentrations of vascular endothelial growth factor (VEGF) and angiopoietin 2 (Ang2) were determined.

### Results

In the ECMO group, MSC subpopulations were higher by 71% compared to the non-ECMO group. Numbers of circulating EPC were not significantly altered. During ECMO, VEGF and

**Data Availability Statement:** All relevant data are within the manuscript and its Supporting Information files.

**Funding:** C.P. & N.R. were supported by research scholarships from the Physician Scientist-Program of the Medical Faculty of the University of Heidelberg. The funder had no role in study design, data collection and analysis, decision to publish, or preparation of the manuscript.

**Competing interests:** The authors have declared that no competing interests exist.

Ang2 serum levels remained unchanged compared to the non-ECMO group (p = 0.16), but Ang2 serum levels in non-survivors of ARDS were significantly increased by 100% (p = 0.02) compared to survivors.

## Conclusions

ECMO support in ARDS is specifically associated with an increased number of circulating MSC, most likely due to enhanced mobilization, but not with a higher numbers of EPC or serum concentrations of VEGF and Ang2.

## Introduction

Extracorporeal membrane oxygenation (ECMO) represents the ultimate life-saving technology for severe forms of acute respiratory distress syndrome (ARDS) [1,2]. Targeted treatment options as well as potent prognostic biomarkers for ARDS patients treated with ECMO, are currently lacking. Considering the importance of endothelial barrier dysfunction in ARDS, current research focuses on the detection of new vascular or endothelium-based targeted therapy options and diagnostic applications in ARDS.

In this respect, endothelial progenitor cells (EPC) and mesenchymal stem cells (MSC) are interesting potential research targets, since they promote vascular regeneration and thereby improve endothelial barrier dysfunction in numerous experimental and pre-clinical models of ARDS [3,4]. The mechanisms by which these cells improve endothelial dysfunction and damage are currently under investigation. It is proposed that these mechanisms include the release of regenerative growth factors, integration into damaged endothelial layers and immune-modulation [5–8]. Mobilization of circulating EPC from the bone marrow is mediated by soluble factors such as vascular endothelial growth factor (VEGF) and angiopoietin-2 (Ang-2), and in ARDS patients in correlation with disease severity [3,9,10].

Furthermore, previous studies suggested that ECMO support might mobilize EPC and MSC into the circulation [11–13]. To this point, no study in ARDS patients has specifically investigated the impact of ECMO on EPC- and MSC mobilization in interaction with potential mobilizing factors like vascular endothelial growth factor (VEGF) and angiopoietin-2 (Ang2). Both EPC and MSC could be useful biomarkers for endothelial dysfunction and regeneration in ARDS while on ECMO-support.

In the present study, we hypothesized that ECMO in ARDS patients upregulates the mobilization of EPC and MSC as well as serum levels of VEGF and Ang2.

## Methods

### Ethics approval and consent to participate

This study was approved by the local Ethics Committee of the Medical Faculty Mannheim of the University of Heidelberg and informed consent was obtained from all study subjects.

### Subjects

Our study has a prospective and observational design. Subjects with ARDS receiving ECMO support ("ECMO group", n = 16) and those without ECMO support ("non-ECMO group", n = 12) were recruited from the intensive care unit (ICU) of the Department of Anaesthesiology and Intensive Care Medicine, University Medical Center Mannheim, University of

Heidelberg within 24 hours after ARDS onset or at admission to the ICU. Subjects were recruited from May 2010 until April 2014. No consecutive subjects were enrolled after the end of April 2014. The subjects included in our study met the diagnostic criteria for ARDS of the American-European Consensus Conference [2]. Disease severity was assessed by the Simplified Acute Physiology Score (SAPS) II [14] and the Therapeutic Intervention Scoring System (TISS) [15]. ECMO was initiated for treatment of hypoxemia ($PaO_2/FiO_2 < 60$) or respiratory acidosis (pH $< 7.2$) despite optimized ventilator settings (tidal volume of 6 mL/g body weight, PEEP set according to the ARDS Network table (AMRA trial) and maximized respiratory rate) [16]. Mortality was defined as death occurring within 28 days after diagnosis. Exclusion criteria were cardiogenic or hemorrhagic shock, chronic obstructive pulmonary disease, absence of mechanical ventilation, and use of angiotensin-converting enzyme inhibitors, activated protein C and hydrocortisone. Clinical data and laboratory findings of each patient were recorded. Blood samples from healthy volunteers from our laboratory staff and their relatives served as age and sex matched healthy controls ("control group", n = 16). This study was approved by the local Ethics Committee of the Medical Faculty Mannheim of the University of Heidelberg and informed consent was obtained from all study subjects or their relatives.

## Blood sampling

Blood (15 mL) was obtained from the central venous catheter of ARDS patients undergoing ECMO before connecting to the ECMO system (day 0). Further blood sampling was performed at day 1, 3 and 7 during ECMO support, directly before and at day 7 and 14 after decannulation or on the day of discharge from the ICU, respectively, if discharged earlier than 14 days after decannulation. Blood samples from ARDS without ECMO support were obtained from the central venous catheter within 24 hours of ARDS diagnosis (day 0), on day 3, 7 and 14 or on the day of discharge from the ICU, respectively, if discharged earlier than 14 days after admission. In healthy controls, 15 mL of blood were collected in tubes containing sodium citrate (0.105 M) as anticoagulant by insertion of a 20-gauge cannula intravenously. The initial 5 mL of blood were discarded to minimize endothelial cell contamination from the puncture wound of the vascular wall. All blood samples were processed within 4 hours after collection. To reflect the respective course of the disease, we averaged all obtained blood samples for each patient of each group. The data generated by this measure are referred to as "averaged"-data in the ongoing text. Laboratory parameters have been obtained at day 0 in both groups. S1 Fig depicts timepoints of blood sampling.

## Flow cytometry

Peripheral blood mononuclear cells (PBMC) were prepared by density gradient centrifugation using Ficoll-Hypaque (Amersham Biosciences, Freiburg, Germany). The expression of cell-surface antigens was determined by immunofluorescence staining as described previously [17]. One hundred microliters of PBMC (containing 3 x $10^6$ cells) were incubated with 20 μL of FcR-blocking reagent (Miltenyi Biotec, Bergisch-Gladbach, Germany) for 10 min to inhibit nonspecific bindings. Thereafter, cells were incubated at 4˚C for 30 min with either 10 μL of PE-conjugated anti-human CD133 monoclonal antibodies (Miltenyi Biotec, Bergisch-Gladbach, Germany), 10 μL of FITC-conjugated anti-human CD34 monoclonal antibodies (BD Biosciences, Heidelberg, Germany), 5 μL of PerCP-conjugated anti-human CD45 monoclonal antibodies and 10 μL of APC-conjugated anti-human CD31 monoclonal antibodies, or 5 μL of PE-conjugated anti-human CD90 monoclonal antibodies (Miltenyi Biotec, Bergisch-Gladbach, Germany), 5 μL of FITC-conjugated anti-human CD29 monoclonal antibodies (BD Biosciences, Heidelberg, Germany), 2 μL of PerCP-conjugated anti-human CD34 monoclonal

antibodies and 5 μL of APC-conjugated anti-human CD73 monoclonal antibodies. Titration experiments have been performed for all antibodies. Isotype-matched immunoglobulin G1 and immunoglobulin G2a antibodies (DakoCytomation, Hamburg, Germany) were used for each patient and measurement as negative controls. The cells were washed three times to remove unbound antibodies and finally resuspended in 400 μL of FACS Cellfix solution (BD Biosciences, Heidelberg, Germany). FACS analysis was performed on a FACSCalibur flow cytometer (BD Biosciences), and the data were analyzed using FlowJo version 7.6.3 software (TreeStar, San Carlos, CA). A minimum of 500,000 events were collected. FACS analysis of each sample was performed in triplicates. The frequency of the expression of surface antigens was determined by a two-dimensional side-scatter/fluorescence dot-plot analysis of the samples after appropriate gating. EPC subpopulations were defined as $CD45^{dim}/CD34^+/CD133^+$ and $CD45^{dim}/CD34^+/CD133^+/CD31^+$ in accordance to established definitions and isolation protocols [18,19]. MSC subpopulations were defined as $CD34^-/CD29^+/CD73^+$, $CD34^-/CD29^+/CD73^+/CD90^+$, $CD34^-/CD73^+/CD90^+$ and $CD34^-/CD29^+/CD90^+$ following published cell surface antigen distribution patterns on MSC [20]. Notably, definitions of MSC are disputed and the populations defined as MSC in our work might also be called "MSC-like" according to the definition of the International Society for Cellular Therapy [21]. EPC and MSC subpopulation numbers are expressed as percentage of total PBMC in each patient or control.

## Enzyme-linked immunosorbent assay

The serum concentrations of vascular endothelial growth factor (VEGF) and Angiopoietin 2 (Ang2) were assessed using enzyme-linked immunosorbent assay kits (R&D Systems, Wiesbaden-Nordenstadt, Germany) in triplicate samples obtained from 1 mL of serum. The enzyme-linked immunosorbent assays were performed according to the manufacturer's instructions.

## Statistical methods

All quantitative data are presented as mean ± standard deviation or as median (range), as appropriate. Both parametric and nonparametric methods were used, as appropriate. All variables were examined for normal and non-Gaussian distribution by the Kolmogorov-Smirnov test. For comparison among normally distributed groups, one-way ANOVA, followed by pairwise multiple comparison (Student-Newman-Keuls method) was used. For non-normally distributed data, the nonparametric Kruskal-Wallis test followed (if necessary) by an all pairwise multiple comparison (Dunn's test) was used. Student's *t*-test and U test were used to compare survival in the ECMO- and non-ECMO group. Logistic regression analysis was performed to predict survival probability from EPC numbers. Correlation analyses (according to Pearson or Spearman) were considered for all target variables that were considered statistically significant. Test results with $p < 0.05$ were considered as statistically significant. All analyses were performed using the SAS system release 9.4 (SAS Institute Inc., NC, USA).

## Results

### Clinical characteristics of the study population

We included 44 study subjects into the analysis. The general characteristics and laboratory findings of the study cohort are presented in Table 1. Subjects in both the ECMO- and the non-ECMO-group were recruited based on a similar ARDS disease severity degree. Thus, there were no significant differences in the score values for TISS and SAPS II between the two patient groups (Table 1). The etiologies within the ECMO group were pneumonias except for one case of pulmonary contusion, in the non-ECMO group all cases were pneumonias.

**Table 1. Characteristics and laboratory findings in all study subjects.**

| | ECMO group | non-ECMO group | | control group |
|---|---|---|---|---|
| | mean | mean | p | mean |
| Male, n [%] | 11 [69] | 7 (58) | - | 12 [75] |
| Female, n [%] | 5 [31] | 5 (42) | - | 4 [25] |
| Survivors, n [%] | 13 [81] | 6 [50] | - | 16 [100] |
| non-survivors, n [%] | 3 [19] | 6 [50] | - | 0 [0] |
| age [years] | 40 ± 16 | 50 ± 13 | 0,09 | 40 ± 16 |
| SAPS2 | 46 ± 14 | 46 ± 9 | 0,76 | ND |
| TISS | 17 ± 4 | 18 ± 6 | 0,40 | ND |
| creatinine [mg/dl] | 1,9 ± 1,1 | 1,7 ± 0,8 | 0,66 | ND |
| hemoglobin [g/dl] | 9,3 ± 0,6 | 10,7 ± 1,9 | 0,08 | ND |
| hematocrit [%] | 29 ± 2,3 | 33 ± 5,2 | **0,04** | ND |
| leukocytes [x$10^9$/l] | 14,5 ± 5,7 | 12,7 ± 3,9 | 0,55 | ND |
| thrombocytes[x$10^9$/l] | 178 ± 81 | 225 ± 126 | 0,40 | ND |
| c-reactive protein [mg/l] | 134 ± 69 | 249 ± 113 | **0,02** | ND |
| procalcitonin [µg/l] | 14,6 ± 14,6 | 4,7 ± 5,7 | **0,03** | ND |

Values for age, SAPS2, TISS, creatinine, hemoglobin, hematocrit, leukocytes, thrombocytes, c-reactive protein and procalcitonin are shown as mean ± SD.

*ND*, no data available; *SAPS II*, Simplified Acute Physiology Score II; *TISS*, Therapeutic Intervention Scoring System (TISS)

Clinical parameters before initiating ECMO in regards to hospitalization, mechanical ventilation, P/F (PaO$_2$/FiO$_2$) ratio and ventilator settings revealed only a significant difference in the P/F ratio (66 ± 33 in the ECMO group vs. 140 ± 85 in the non-ECMO group, p = 0.0008), but not in the other parameters (Table 2)

When looking at the mean laboratory parameters, the blood cell counts showed no significant differences between both patient groups (Table 1). But serum levels of CRP were significantly increased by 86% in the non-ECMO group. In addition, supplementary Table 2 shows SPAS2, TISS, CRP and Procalcitonin (Pct) in both the ECMO–and the non-ECMO group at day 0. Procalcitonin at day 0 was significantly higher in the ECMO-group (18,6 ± 17,2 µg/l vs. 5,3 ± 7,83 µg/l, p = 0,02).

**Table 2. Clinical parameters before initiating ECMO in regards to hospitalization, mechanical ventilation, P/F ratio and ventilator settings.**

| | ECMO group | non-ECMO group | |
|---|---|---|---|
| | mean | mean | p |
| **Hospitalization** | | | |
| hospital days before MV, *d* | 2.9 ± 3.2 | 2.17 ± 1.46 | 0.25 |
| **Mechanical Ventilation** | | | |
| days of MV before ECMO, *d* | 2.0 ± 1.4 | - | - |
| days of MV before admission, *d* | - | 3.4 ± 8.7 | - |
| **P/F ratio** | | | |
| paO$_2$/FiO$_2$ ratio before ECMO/at day 1 | 66 ± 33 | 140 ± 85 | **0.0008** |
| **Ventilator Settings** | | | |
| PIP before ECMO/at day 1, *cm H$_2$O* | 33 ± 4.6 | 32 ± 3.1 | 0.24 |
| PEEP before ECMO/at day 1, *cm H$_2$O* | 17 ± 3.2 | 17 ± 5.3 | 0.5 |
| F$_{iO2}$ before ECMO/at day 1 | 0.87 ± 0.2 | 0.85 ± 0.17 | 0.36 |

*ECMO*, extracorporeal membrane oxygenation; *F$_{iO2}$*, fraction of inspired oxygen; *MV*, mechanical ventilation; *PEEP*, positive end-expiratory pressure; PIP, peak inspiratory pressure.

## Subpopulations of EPC and MSC in disease course

The numbers of all EPC subpopulations were significantly increased in the ECMO group and the non-ECMO group compared to the control group (Fig 1A). In addition, the numbers of the MSC subpopulations CD34⁻/CD73⁺/CD90⁺ and CD34⁻/CD73⁺/CD29⁺/CD90⁺ in the ECMO group were significantly increased compared to the non-ECMO group (e.g. by 71% for MSC CD34⁻/CD73⁺/CD29⁺/CD90⁺) (Fig 1A). In the non-ECMO group the numbers of MSC CD34⁻/ CD73⁺/CD90⁺ showed a significant increase compared to the control group (Fig 1A). For EPC, we detected slightly increased numbers in the ECMO group compared to the non-ECMO group, yet these results were not significant (EPC CD45$^{dim}$/CD34⁺/CD133⁺: p = 0.26; EPC CD45$^{dim}$/CD34⁺/CD133⁺/CD31⁺: p = 0.28) (Fig 1A). When comparing the numbers of EPC and MSC subpopulations at the different time points in the ECMO and the non-ECMO group, EPC-subpopulations were increased at day 7 after initiation of ECMO support in the ECMO group compared to day 7 in the non-ECMO group. MSC subpopulations were significantly increased at day 0 in the ECMO-group compared to the non-ECMO group (Fig 1B–1E).

## Serum levels of mobilizing factors

Serum levels of Ang2 and VEGF were significantly increased in both ARDS groups compared to the control group (Fig 2A). The averaged serum levels of VEGF- and Ang2 did not differ between the ECMO- and the non-ECMO group (Fig 2A). However, comparing each time point of blood sampling in each group revealed a significant decrease of VEGF serum levels at day 1 and day 3 after initiation of ECMO support and a significant decrease of Ang2 serum levels at day 14 in the non-ECMO group (Fig 2B). Ang2 serum levels neither correlated with MSC nor with EPC numbers (data not shown). VEGF serum levels correlated with EPC subpopulations (S1 Table) but not with MSC subpopulations (data not shown).

## Survival analysis regarding EPC, MSC and the mobilizing factors

Survivors of all ARDS subjects had slightly increased numbers of EPC and MSC in our study. However, these findings were statistically not significant (Fig 3A). Serum levels of Ang2 in ARDS subjects were significantly increased in non-survivors compared to survivors by 100% (Fig 3C). For VEGF, no significant difference was found between survivors and non-survivors (p = 0.31) (Fig 3B). When comparing the survivors and non-survivors of the ECMO group to the survivors and non-survivors of the non-ECMO group, no significant result was found neither for the different subpopulations of EPC and MSC nor for the mobilizing factors VEGF and Ang2 (S1 Fig).

## Discussion

In the present study, we examined the impact of ECMO on the mobilization of EPC and MSC in ARDS patients. This study is the first that demonstrates that ECMO support in ARDS is associated with an increased number of circulating MSC, while the number of circulating EPC was not significantly different among groups. Neither VEGF nor Ang2 serum levels showed a significant association with ECMO support in the clinical course of ARDS, but in the first days after initiation of ECMO support, VEGF serum levels declined significantly. ARDS non-survivors showed increased levels of Ang2 compared to survivors, while there was no difference with VEGF levels between survivors and non-survivors.

The mobilization of EPC and MSC is differentially regulated in several diseases with marked endothelial dysfunction or systemic inflammation such as sepsis and ARDS [9,17,22]. Animal models of these diseases and experimental studies have demonstrated that EPC and

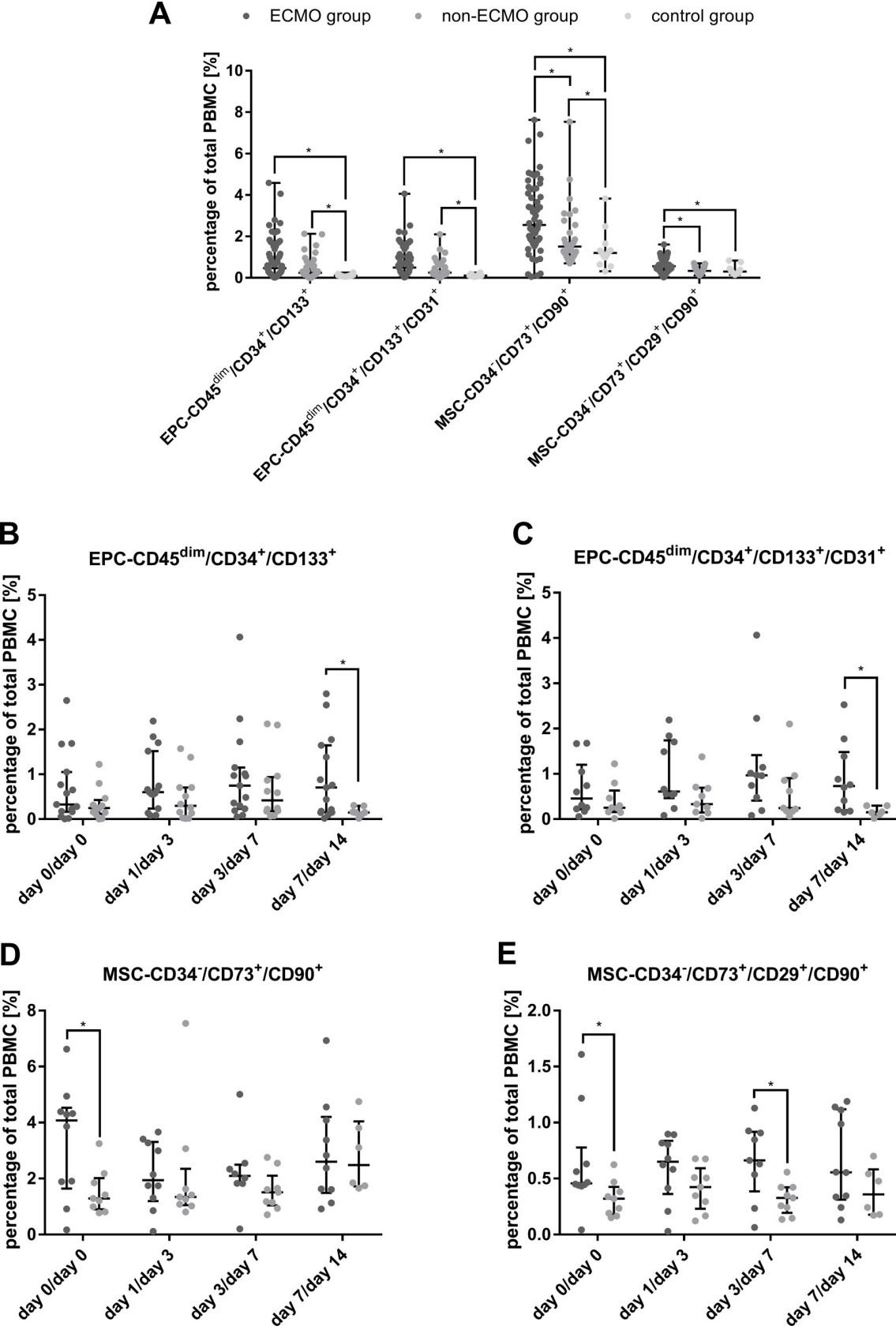

**Fig 1. Upregulation of EPC and MSC populations in study subjects.** (A) The averaged numbers of endothelial progenitor cell (EPC) and mesenchymal stem cell (MSC) subpopulations are shown for the ECMO, the non-ECMO and the control group. In panel B-E, the numbers of the subpopulations of EPC and MSC are displayed for the different time points in the disease course of the ECMO and the non-ECMO group (in the ECMO group at day 0, day 1, day 3 and day 7; in the non-ECMO group at day 0, day 3, day 7 and day 14). *marks a significant difference (p<0.05).

MSC beneficially influence endothelial function and promote regeneration based on proangiogenic signaling [5,6,23,24]. In addition, previous studies suggest that ECMO support might increase the mobilization of EPC and MSC into the circulation [11–13]. Although ARDS patients in our study exhibited similar disease severity assessed by SAPS2 and TISS, subjects in the ECMO group showed increased numbers of the MSC subpopulations CD34⁻/CD73⁺/CD90⁺ and CD34⁻/CD73⁺/CD29⁺/CD90⁺ in the disease course compared to the non-ECMO group. The comparable disease severity scores in both groups might have also minimized a presumably confounding impact on the mobilization of both stem cell populations, which might arise from heterogeneity of disease etiologies and disease severity differences. Therefore, the observed changes in MSC numbers might be essentially attributable to the ECMO support. However, we cannot exclude certain confounding influences by disease etiology or differences in inflammatory marker levels between both ARDS groups.

Soluble factors such as VEGF and Ang-2 mediate the mobilization from bone marrow through the activation of metalloproteinases and upregulation of adhesion molecules [3,25–27]. We have previously demonstrated that plasma levels of VEGF and Ang-2 are increased in septic patients and correlate with the number of EPC [28,29]. Similar findings were shown for Ang-2 in ARDS [30,31]. With regards to VEGF, a correlation between increased plasma VEGF levels and severity of Multi-Organ-Dysfunction-Syndrome (MODS) has been reported by several groups [32,33]. The role of VEGF in ARDS is still matter of debate, whether it contributes to the pathogenesis by increasing pulmonary vascular permeability, or acts as a cellular growth factor thereby inducing vascular regeneration [34]. An association between MSC and EPC numbers and VEGF serum levels has been described before [17]. In our study, VEGF serum levels were decreased during the first three days after initiation of ECMO support. However, overall mean VEGF serum levels in the ECMO group were not significantly altered compared to the non-ECMO group, but all EPC subpopulations correlated with VEGF serum levels. Since VEGF, as an important and potent mobilizer of EPC [26,35], remained largely unaffected by ECMO support, this might explain, why EPC numbers in the ECMO group showed no significant changes. VEGF also plays an important role for MSC biology [36,37], but in our study, MSC subpopulation numbers did not correlate with VEGF serum levels, indicating that VEGF might not be responsible for the increase in MSC numbers during ECMO.

Besides VEGF, Ang2 also plays a distinct role in ARDS pathophysiology. It has been proposed that Ang2 might act as an adverse player, since its serum levels correlate with mortality in ARDS [38]. Our results demonstrate that both ARDS groups had higher Ang2 serum levels compared to the control group. In addition, Ang2 levels were significantly higher in non-survivors compared to survivors and were associated with an increased probability to die from ARDS. In the course of ARDS in our study, Ang2 serum levels were significantly decreased after two weeks in the non-ECMO group, while Ang2 serum levels remained stable in the ECMO group. Also, Ang2 serum levels neither correlated with MSC nor with EPC numbers. Therefore, the increase in MSC numbers in the ECMO group seems not to be associated with Ang2. However, since Ang2 has been shown to improve MSC functions like migration, induction of angiogenesis and the secretion of paracrine factors [27,39], the observed maintenance of Ang2 serum levels during ECMO support might appear favorable regarding the regenerative potential of MSC.

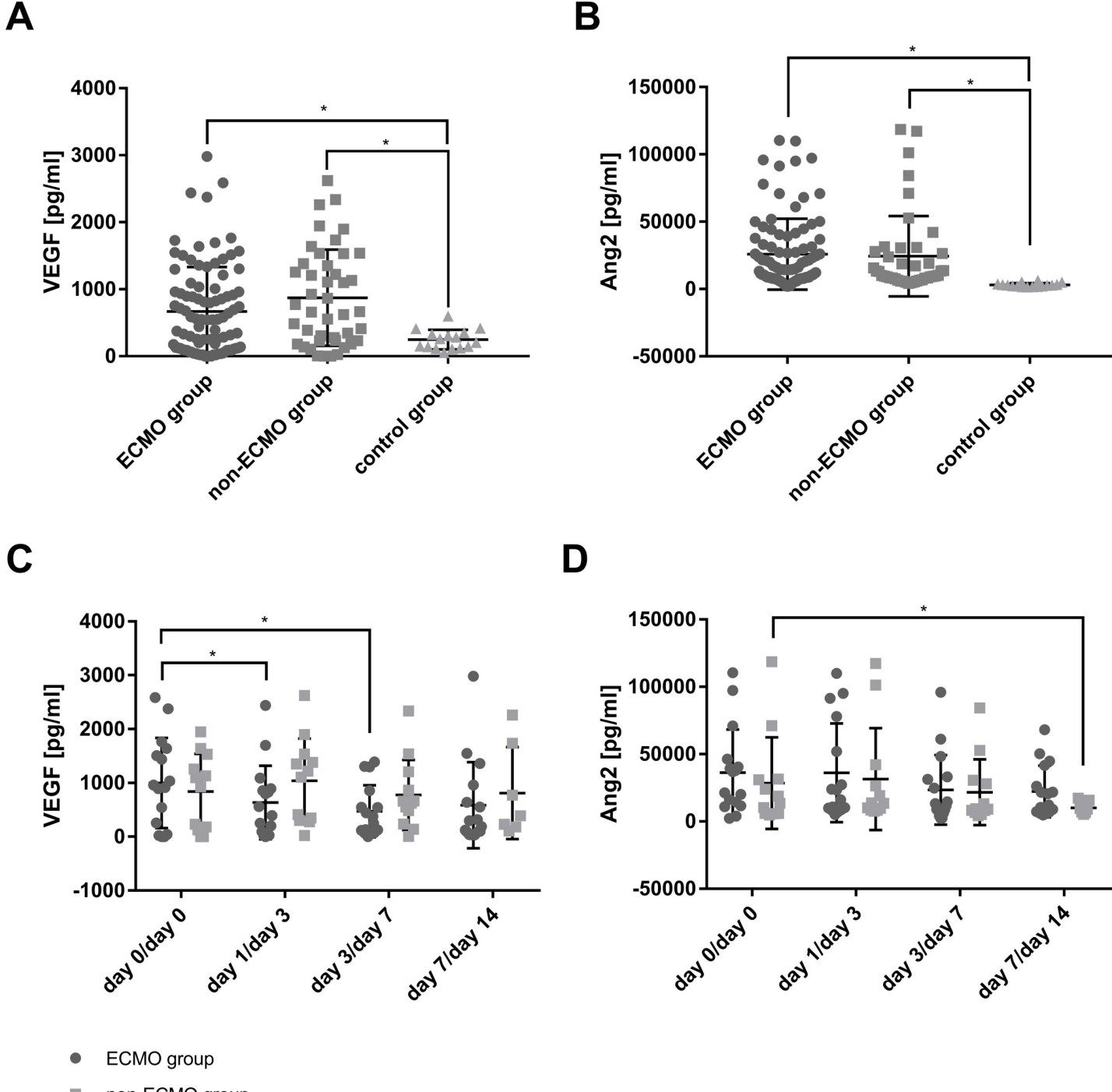

**Fig 2. Serum levels of mobilizing factors.** The averaged levels of vascular endothelial growth factor (VEGF) and angiopoietin 2 (Ang2) (panel A and B) and for each timepoint of blood sampling (panel C and D) (in the ECMO group at day 0, day 1, day 3 and day 7; in the non-ECMO group at day 0, day 3, day 7 and day 14) are displayed. *marks a significant difference (p<0.05).

To interpret our results within a clinical context, we have to discuss the limitations of our study. Although our study has a prospective design, it is, however, an observational and not an interventional study, which has an impact on data interpretation. We were not in control of

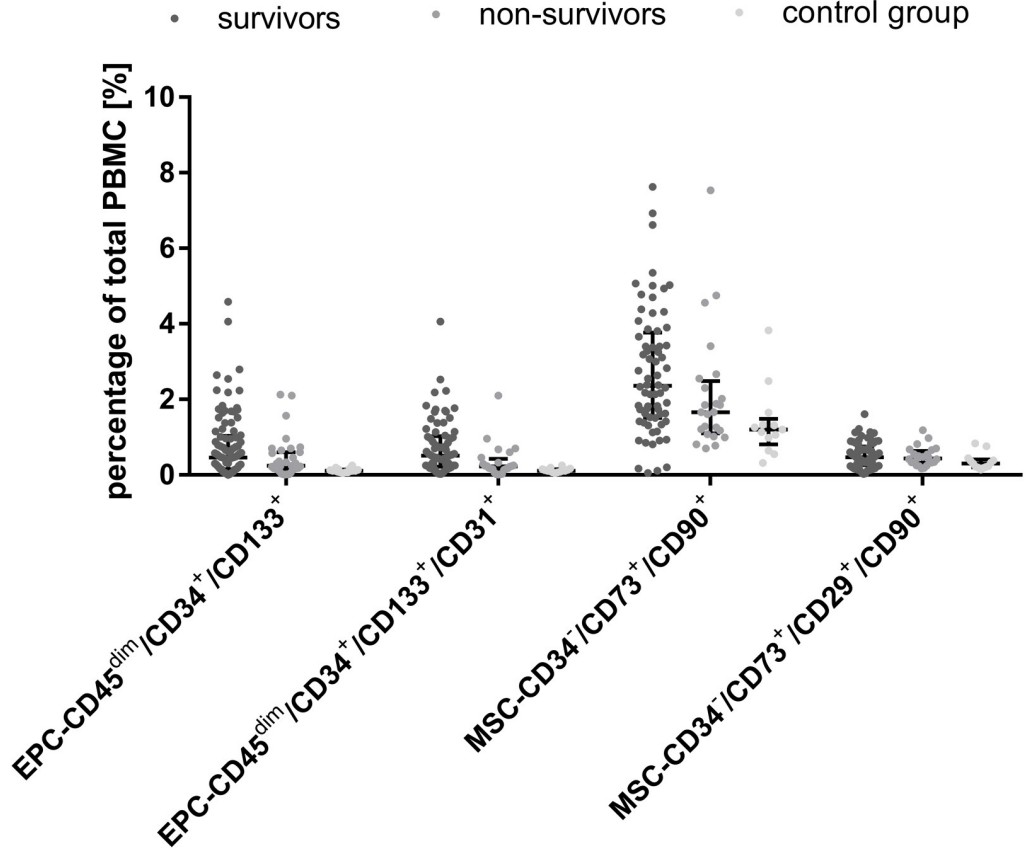

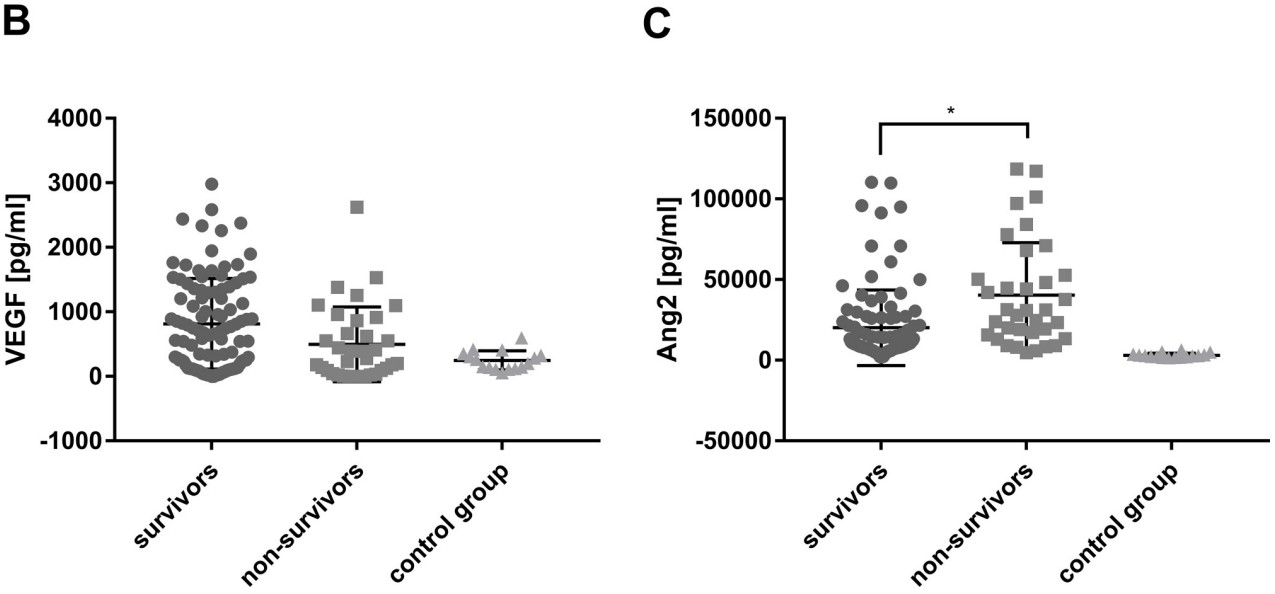

**Fig 3. Association between survival and numbers of EPC and MSC as well as mobilizing factors.** Numbers of endothelial progenitor cell (EPC) and mesenchymal stem cell (MSC) subpopulations (panel A) as well as serum levels of vascular endothelial growth factor (VEGF) (panel B) and angiopoietin 2 (Ang2) (panel C) are shown for ARDS survivors and non-survivors. * marks a significant difference (p<0.05).

certain confounders and can mainly observe associations and not deduce cause and effect. Thus, the conclusions drawn from our results on the impact of ECMO on EPC, MSC, VEGF and Ang2 remain mainly associative. Also, it is not clear yet, whether the increased MSC numbers in ARDS patients actually assist in healing the lung injury or whether they are only a biomarker of disease severity. We did not clearly show a statistically significant survival benefit in ARDS patients with increased MSC numbers. Thus, we also cannot exclude that the increase in MSCs might also be associated with a deterioration of the lung injury. However, this latter option seems unlikely in the light of current promising clinical trials on MSC applications in ARDS [40]. Besides VEGF and Ang2, our study does not include other factors associated with ARDS and ECMO like inflammatory cytokines, which might have an influence on EPC and MSC mobilization. The underlying disease in our patients–pneumonia–might have also influenced the mobilization differentially, depending on the severity or the microbiological agent causing the pneumonia. Furthermore, it must be discussed that the need for blood products or volume support during ECMO might have confounded serum levels and cell counts in patient samples, which we did not assess systematically. Furthermore, we want to acknowledge that our observations relative to ECMO may not relate to all ECMO devices and strategies. Also, the impact of ventilator settings is not known yet. In addition, we did not perform functional analyses with the identified EPC- and MSC subgroups in our study. Thus, we can only assume that they behave according to already published data on the biology of these cells. It should also be noted that subjects presented in our study represent a small number of individuals and–suitable for this pilot data–form a convenience sample.

Current research focuses on stem cell-based treatment strategies, and especially the application of MSC has been the object of many clinical studies due to its immune privileged characteristics. Since MSC do not express human leucocyte antigens (HLA) on their surface, there is no immunological risk for allogenic transplantation. Several clinical studies are currently being performed to investigate the efficacy and compatibility of MSC application in ARDS and other diseases (see https://clinicaltrials.gov). In the START-study (NCT01775774) [40], which examined the safety of exogenously applied MSC in ARDS patients, the application of MSC was well tolerated, so that the group has proceeded to a phase II-trial with a primary focus on safety and secondary outcomes including respiratory, systemic, and biological endpoints. Special interest is also given to the STELLAR-study (NCT02112500), which examines as a phase II-trial the efficacy and safety of administered MSC in ARDS patients and the REALIST-study (NCT03042143), which first evaluates the safety of MSC application in a phase I-trial and consecutively the efficacy in a phase II-trial. The results of these studies are eagerly awaited and will give important evidence about the therapeutic potential of MSC-based treatment strategies. However, the application of MSC in patients treated with ECMO needs further thorough clinical investigation. Recently, Millar et al. demonstrated a rapid decline in oxygenator performance in an ex-vivo model of ECMO [41]. Another group evaluated the viability and activity of MSC in ex vivo circulation conditions with and without an oxygenator [42]. The viability and activity of MSC decreased significantly when the oxygenator was used, mainly due to lysis and to the nonphysiologic condition itself [42]. Therefore, further studies need to evaluate MCS viability and activity in in-vivo models of ECMO.

Aside from MSC, numerous studies have demonstrated a positive impact of EPC on pulmonary vascular growth and regeneration [3,43]. However, some studies show conflicting results or fail to demonstrate an impact of bone marrow-derived EPC on vascular regeneration in

acute lung injury and other conditions [44,45]. This discrepancy might most probably be attributable to differences in cell populations applied and in cell isolation techniques. Therefore, the best population and mode of application has still to be identified.

In summary, our results indicate that ECMO support in ARDS is associated with increased MSC mobilization. This finding confirms the results of both Hoesli and Lehle et al. who demonstrated the isolation of progenitor cells on ECMO membranes and within the ECMO circuit [11,13]. Our results also confirm those of Bui et al. [46], who suggested, that ECMO increases the number of peripheral progenitor and stem cells. However, the underlying diseases and patient age in their study population were heterogeneous. In this current study, we now specifically focused on ARDS as a single disease etiology and investigated patients with similar disease severity [46]. This probably has minimized a presumably confounding impact by disease etiology and severity on the mobilization of stem cells. An association between MSC mobilization and VEGF and Ang2 could not be demonstrated, so that the underlying mechanisms of MSC mobilization during ECMO in ARDS remain unknown. Future studies might want to put a focus on other factors, like CXC-motive-chemokinereceptor 4 (CXCR-4), insulin-like growth factor 1 (IGF-1), stromal-derived factor 1 (SDF-1) and T-cell derived interferone-γ (IFN-v)–which have been shown to mobilize MSC from the bone marrow [22,47] and might thus be involved in an ECMO-induced increase in MSC numbers in ARDS. Furthermore, the impact of ECMO support on EPC mobilization in ARDS was less pronounced compared to MSC. These results raise the question whether an increase of EPC, MSC, VEGF or Ang2 could beneficially influence the clinical course of ARDS patients undergoing ECMO, which needs to be addressed in future studies. An increase of either MSC or EPC could be accomplished by stimulation of endogenous mobilization or by exogeneous cell transplantation.

## Supporting information

**S1 Fig. Timeline-flowchart depicting timepoints of blood sampling.**
(DOCX)

**S2 Fig. Association of survival with numbers of EPC and MSC as well as mobilizing factors in the ECMO-dependent and the ECMO-independent group.** Numbers of endothelial progenitor cell (EPC) and mesenchymal stem cell (MSC) subpopulations (panel A) as well as serum levels of vascular endothelial growth factor (VEGF) (panel B) and angiopoietin 2 (Ang2) (panel C) are shown for the survivors and non-survivors in the ECMO-dependent and the ECMO-independent group. *marks a significant difference (p<0.05).
(DOCX)

**S1 Table. Shows Pearson correlation coefficients between EPC subpopulations and VEGF serum levels at day 0 and in the disease course.**
(DOCX)

**S2 Table. Characteristics and laboratory findings in ARDS patients at day 0.**
(DOCX)

**S1 Rawdata.**
(XLSX)

## Author Contributions

**Conceptualization:** Christian Patry, Thalia Doniga, Christel Weiss, Burkhard Tönshoff, Armin Kalenka, Benito Yard, Jörg Krebs, Thomas Schaible, Grietje Beck, Neysan Rafat.

**Data curation:** Neysan Rafat.

**Formal analysis:** Christian Patry, Thalia Doniga, Franziska Lenz, Tim Viergutz, Christel Weiss, Burkhard Tönshoff, Armin Kalenka, Benito Yard, Jörg Krebs, Thomas Schaible, Grietje Beck, Neysan Rafat.

**Investigation:** Franziska Lenz, Tim Viergutz.

**Supervision:** Neysan Rafat.

**Validation:** Neysan Rafat.

**Writing – original draft:** Christian Patry, Thalia Doniga.

**Writing – review & editing:** Neysan Rafat.

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
