## [Decision Letter · Decision Letter 0]

25 Sep 2019

PONE-D-19-21924

Increased mobilization of mesenchymal stem cells in patients with acute respiratory distress syndrome undergoing extracorporeal membrane oxygenation

PLOS ONE

Dear Dr. Patry,,

Thank you for submitting your manuscript to PLOS ONE. After careful consideration, we feel that it has merit but does not fully meet PLOS ONE’s publication criteria as it currently stands. Therefore, we invite you to submit a revised version of the manuscript that addresses the points raised during the review process.

We would appreciate receiving your revised manuscript by January 31, 2020. To enhance the reproducibility of your results, we recommend that if applicable you deposit your laboratory protocols in protocols.io, where a protocol can be assigned its own identifier (DOI) such that it can be cited independently in the future. For instructions see: http://journals.plos.org/plosone/s/submission-guidelines#loc-laboratory-protocols

We look forward to receiving your revised manuscript.

Kind regards,

You-Yang Zhao

Academic Editor

PLOS ONE

Journal Requirements:

Additional Editor Comments (if provided):

Reviewers' comments:

Reviewer's Responses to Questions

**Comments to the Author**

1. Is the manuscript technically sound, and do the data support the conclusions?

Reviewer #1: Yes

Reviewer #2: Yes

2. Has the statistical analysis been performed appropriately and rigorously? 

Reviewer #1: I Don't Know

Reviewer #2: I Don't Know

3. Have the authors made all data underlying the findings in their manuscript fully available?

Reviewer #1: Yes

Reviewer #2: Yes

4. Is the manuscript presented in an intelligible fashion and written in standard English?

Reviewer #1: Yes

Reviewer #2: Yes

5. Review Comments to the Author

Reviewer #1: MAJOR

1. The contribution of bone marrow-derived cells to repair from acute inflammatory injury has shown to be limited and/or widely variable in experimental studies of lung injury (e.g. in the studies reviewed by the senior author in his AJRCMB review in 2013). Authors should reconcile/discuss these diverse findings in the Discussion.

2. From the current data, it is not clear whether the increased number of MSCs contribute to the lung injury and/or help to repair the lung injury, particularly given that there are no survival differences associated with cell mobilization. It would be useful to include further studies that assess the impact of the increased cell types using in vitro and/or in vivo models of inflammatory endothelial/lung injury. If no further data is added, this potential caveat should be clearly explained in the Discussion.

3. There is no data to investigate how ECMO might trigger MSC mobilization. At the very least, suggestions for possible future studies that would investigate regulatory mechanisms should be included in the Discussion.

4. What “blood products” and/or “volume support” were given to the patients? These should be reported, given that they could directly alter the main outcome variables being measured.

MINOR

1. Some figures are box plots, while other figures are scatter plots. It would be better for all figures to be scatter plots of individual data points.

2. It would be useful to provide a timeline figure showing when blood samples were taken in each of the groups and with respect to ECMO. Related to this, the x axis labels are confusing in Figure 1B-E and 2C and 2D. It seems to me that comparing different timepoints for the groups could be inappropriate (e.g. day 1 for ECMO vs day 3 for non-ECMO); wouldn’t it be better to compare the groups at the same time points (e.g. day 3 ECMO versus day 3 non-ECMO)? Further, it would be helpful throughout the text if the actual timepoints were stated instead of referring to “initial” or “early” or “later” times.

3. “previous studies suggest an association between ECMO support and increased numbers of EPC and MSC in the circulation [11–13].” Authors should mention how the findings of these referenced studies compare with the current data. It should also be clear in the Introduction/Discussion how the current study adds to these referenced findings.

4. Supp Fig 1A-C: the legend suggests there are some significant differences marked by a * but the figure and text state there are no significant differences in survival.

Reviewer #2: Patry et al. present an interesting observational study of ARDS patients supported with ECMO. The study focuses on the effect of ECMO on circulating populations of EPCs and MSCs. This is an interesting and evolving area of research and as such is likely to be of interest to a broad audience in the critical care medicine field. The authors clearly detail that they have obtained the required ethical approvals for their study.

In general, the manuscript is well written and has a high level of scientific English. The manuscript is appropriately set out and the accompanying figures are well presented and informative. The supplementary information included is helpful. I do have a few major concerns and some minor concerns which I have detailed below.

Major concerns.

1. In several sections of the manuscript the authors make reference to the similarity of disease severity between groups, particularly in an early section of the discussion. While this may be true of the mean SAPS II scores, PF ratios are very different at baseline (prior to ECMO). This has two implications, the first is that given PF ratio is a SAPS II variable are the non-ECMO patients acquiring extra points in another domain? Secondly, these marked differences in PF and CRP/PCT would suggest these are not the equivalent groups even before ECMO is initiated. Given these points the following statement seems particularly overstated

''The comparable disease severity scores in both groups might have also minimized a presumably confounding impact on the mobilization of both stem cell populations, which might arise from heterogeneity of disease etiologies and disease severity differences. Therefore, the observed changes in MSC numbers might be rather attributable to the ECMO support itself instead of being triggered by ARDS disease severity differences between both ARDS groups.''

2. Statistical analysis. Has an attempt been made to correct for multiple comparisons? Is pairwise comparison at each time interval the most efficient means of analysis in this design. Given these are essentially repeated measures, would repeated-measures ANOVA or a mixed model not be preferential.

3. Figure 1. I am a little confused as to the content of panel A. Is this at a specific time-point or is it some form of composite? Does it include day 0?

4. While highlighting the potential benefits of cell therapy in ARDS, in particular MSCs, the authors have failed to address important and emerging evidence of the harms of MSCs in ECMO, see:

Millar et al. Thorax 2019 10.1136/thoraxjnl-2017-211439

and

Cho et al. ASAIO J 2019 10.1097/MAT.000000000000102519

Evidence from complex large animal models which substantiate these claims have been presented in abstract form at a number of meetings.

Minor concerns

1. On line 23 oxygenation index (OI) appears to have been confused with P/F ratio.

2. This is of more minor importance given definitions are disputed, however the cell population characterised as MSCs in this paper are MSC-like when using the ISCT definition (10.1080/14653240600855905).

3. The Lehle et al. study has been used incorrectly in varying parts of the manuscript to support the presence of circulating MSCs during ECMO. This study had no control group and in fact the study was based on cells adherent to the membrane oxygenator.

4. Was this all VV ECMO? It would be beneficial to also have some idea of the differences in ventilatory settings between patients on or nor on ECMO. Perhaps this could be included as supplementary table.

5. The mortality difference is marked between groups, where there any patients in the ARDS non-ECMO group who were refused ECMO?

6. PLOS authors have the option to publish the peer review history of their article (what does this mean?). If published, this will include your full peer review and any attached files.

Reviewer #1: Yes: Colin E. Evans

Reviewer #2: No

---

## [Author Response · Author response to Decision Letter 0]

5 Dec 2019

Point-by-point response to the reviewer

We thank the reviewers for reviewing our manuscript and their thoughtful comments.

Reviewer 1

Major critiques:

1.) The contribution of bone marrow-derived cells to repair from acute inflammatory injury has shown to be limited and/or widely variable in experimental studies of lung injury (e.g. in the studies reviewed by the senior author in his AJRCMB review in 2013). Authors should reconcile/discuss these diverse findings in the Discussion.

To 1.) We thank the reviewer for his suggestion. We now have commented on studies regarding EPC-based regeneration in experimental models of lung injury. 

 Page 17, para 2: 

“Aside from MSC, numerous studies have demonstrated a positive impact of EPC on pulmonary vascular growth and regeneration [3,42]. However, some studies show conflicting results or fail to demonstrate an impact of bone marrow-derived EPC on vascular regeneration in acute lung injury and other conditions [43,44]. This discrepancy might most probably be attributable to differences in cell populations applied and in cell isolation techniques. Therefore, the best population and mode of application has still to be identified.” 

2.) From the current data, it is not clear whether the increased number of MSCs contribute to the lung injury and/or help to repair the lung injury, particularly given that there are no survival differences associated with cell mobilization. It would be useful to include further studies that assess the impact of the increased cell types using in vitro and/or in vivo models of inflammatory endothelial/lung injury. If no further data is added, this potential caveat should be clearly explained in the Discussion.

To 2.) We did already lay out in the discussion that our results remain mainly associative. As suggested by the reviewer, we have added now a new comment on the question of actual survival benefit by increased MSCs to the limitations section. We hope the reviewer agrees to this.

 Page 15, para 3:

 “Also, it is not clear yet, whether the increased MSC numbers in ARDS patients actually assist in healing the lung injury or whether they are only a biomarker of disease severity. We did not clearly show a statistically significant survival benefit in ARDS patients with increased MSC numbers. Thus, we also cannot exclude that the increase in MSCs might also be associated with a deterioration of the lung injury. However, this latter option seems unlikely in the light of current promising clinical trials on MSC applications in ARDS [40].”

3.) There is no data to investigate how ECMO might trigger MSC mobilization. At the very least, suggestions for possible future studies that would investigate regulatory mechanisms should be included in the Discussion.

To 3.) According to the reviewers’ suggestion, we have commented on possible mechanisms, which might be involved in MSC release during ECMO in ARDS.

 Page 18, para 1:

 “An association between MSC mobilization and VEGF and Ang2 could not be demonstrated, so that the underlying mechanisms of MSC mobilization during ECMO in ARDS remain unknown. Future studies might want to put a focus on other factors, like CXC-motive-chemokinereceptor 4 (CXCR-4), insulin-like growth factor 1 (IGF-1), stromal-derived factor 1 (SDF-1) and T-cell derived interferone-y (IFN-v) – which have been shown to mobilize MSC from the bone marrow [22,46] and might thus be involved in an ECMO-induced increase in MSC numbers in ARDS.”

4.) What “blood products” and/or “volume support” were given to the patients? These should be reported, given that they could directly alter the main outcome variables being measured.

To 4.) We agree with the reviewer that blood products and/or volume support can alter the outcome variables. However, we did not systematically assess frequency of blood product application or volume support in the patients of our study and we missed to address this as a limiting factor in our discussion. We now added a respective section into the limitations, which reads as follows:

 Page 16, para 1:

 “Furthermore, it must be discussed, that the need for blood products or volume support during ECMO might have confounded serum levels and cell counts in patient samples, which we did not assess systematically. 

Minor critiques:

5.) Some figures are box plots, while other figures are scatter plots. It would be better for all figures to be scatter plots of individual data points.

To 5.) We have now provided a new Figure 2 and 3 showing all figures with individual data points. 

6.) It would be useful to provide a timeline figure showing when blood samples were taken in each of the groups and with respect to ECMO. Related to this, the x axis labels are confusing in Figure 1B-E and 2C and 2D. It seems to me that comparing different timepoints for the groups could be inappropriate (e.g. day 1 for ECMO vs day 3 for non-ECMO); wouldn’t it be better to compare the groups at the same time points (e.g. day 3 ECMO versus day 3 non-ECMO)? Further, it would be helpful throughout the text if the actual timepoints were stated instead of referring to “initial” or “early” or “later” times.

To 6.) We thank the reviewer for his thoughtful comments. The reviewer is correct, that comparing different timepoints in different groups might seem inappropriate here. 

 However, the respective timepoints in each of the two groups – even if called the same, like “day 3 ECMO group” and “day 3 ARDS group” - are not necessarily the same timepoints in chronological terms. This is because we defined “day 0” in the ARDS group as the initial timepoint of ARDS diagnosis. However, in the ECMO-group “day 0” is the day before start of ECMO which is not necessarily the day at which the diagnosis of ARDS might have arisen. This is why it seemed most appropriate to us, to compare timepoints within the two groups, with the smallest chronological interspace between them. This means “day0/day0”, “day1/day3”, “day3/day7” and “day7/day14”. Thereby, we wanted to reflect the course of disease (in the ECMO-group starting with the initiation of EMCO, in the ARDS group starting with the diagnosis of ARDS) rather than compare timepoints specifically.

According to the reviewers‘ suggestion, we have omitted phrases like “earlier”, “later” and “initial” in the manuscript and instead referred to the actual timepoints of blood sampling.

The respective changes now read as follows:

Page 10, Para 2:

“When comparing the numbers of EPC and MSC subpopulations at the different time points in the ECMO and the non-ECMO group, EPC-subpopulations were increased at day 7 after initiation of ECMO support in the ECMO group compared to day 7 in the non-ECMO group. MSC subpopulations were significantly increased at day 0 in the ECMO-group compared to the non-ECMO group (Fig 1B-E).”

Page 11, Para 3:

However, comparing each time point of blood sampling in each group revealed a significant decrease of VEGF serum levels at day 1 and day 3 after initiation of ECMO support and a significant decrease of Ang2 serum levels at day 14 in the non-ECMO group (Fig 2B).

Page 14, Para 2:

In our study, VEGF serum levels were decreased during the first three days after initiation of ECMO support.

Page 15, Para 2:

In the course of ARDS in our study, Ang2 serum levels were significantly decreased after two weeks in the non-ECMO group, while Ang2 serum levels remained stable in the ECMO group.

Furthermore, we have now provided a timeline flowchart showing when blood samples were taken in each group in a new Figure 4.

7.) “Previous studies suggest an association between ECMO support and increased numbers of EPC and MSC in the circulation [11–13].” Authors should mention how the findings of these referenced studies compare with the current data. It should also be clear in the Introduction/Discussion how the current study adds to these referenced findings.

To 7.) According to the reviewers’ suggestion we added the following statement in the conclusion section of our discussion.

 Page 17, para 3:

 “This finding confirms the results of both Hoesli and Lehle et al. who demonstrated the isolation of progenitor cells on ECMO membranes and within the ECMO circuit [11,13]. Our results also confirm those of Bui et al. [45], who suggested, that ECMO increases the number of peripheral progenitor and stem cells. However, the underlying diseases and patient age in their study population were heterogeneous. In this current study, we now specifically focused on ARDS as a single disease etiology and investigated patients with similar disease severity [45]. This probably has minimized a presumably confounding impact by disease etiology and severity on the mobilization of stem cells.” 

8.) Supp Fig 1A-C: the legend suggests there are some significant differences marked by a * but the figure and text state there are no significant differences in survival.

To 8.) Thank you for indicating this error, which was due to a “copy and paste” approach. We deleted the significance statement in the respective figure legend. Please, kindly excuse the inconvenience.

Reviewer 2

Major critiques:

1.) In several sections of the manuscript the authors make reference to the similarity of disease severity between groups, particularly in an early section of the discussion. While this may be true of the mean SAPS II scores, PF ratios are very different at baseline (prior to ECMO). This has two implications, the first is that given PF ratio is a SAPS II variable are the non-ECMO patients acquiring extra points in another domain? Secondly, these marked differences in PF and CRP/PCT would suggest these are not the equivalent groups even before ECMO is initiated. Given these points the following statement seems particularly overstated

''The comparable disease severity scores in both groups might have also minimized a presumably confounding impact on the mobilization of both stem cell populations, which might arise from heterogeneity of disease etiologies and disease severity differences. Therefore, the observed changes in MSC numbers might be rather attributable to the ECMO support itself instead of being triggered by ARDS disease severity differences between both ARDS groups.''

To 1.) We thank the reviewer for his remark. We agree with the reviewer, that certain disease severity differences might have had an impact on MSC numbers in both ARDS groups. We have added a new table to the supplementary files (Suppl. Table 2), which shows SAPS, TISS, CRP and Pct at day 0 in both groups. Both disease severity scores do not differ between the ECMO- and the non-ECMO group but Pct-Values were higher in the ECMO-group. 

 We have added this information in the Results section, which now reads as follows:

 Page 9, para 2: 

 “In addition, supplementary Table 2 shows SPAS2, TISS, CRP and Pct in both the ECMO – and the non-ECMO group at day 0. Procalcitonin at day 0 was significantly higher in the ECMO-group (18,6 ± 17,2 µg/l vs. 5,3 ± 7,83 µg/l, p=0,02).”

 This finding suggests a certain impact on different disease etiologies or severities, not assessed by SAPS and/or TISS. We thus have changed our statement in the discussions section as follows:

 “''The comparable disease severity scores in both groups might have also minimized a presumably confounding impact on the mobilization of both stem cell populations, which might arise from heterogeneity of disease etiologies and disease severity differences. Therefore, the observed changes in MSC numbers might be essentially attributable to the ECMO support, however, we cannot exclude certain confounding influences by disease etiology or differences in inflammatory marker levels between both ARDS groups.”

2.) Statistical analysis. Has an attempt been made to correct for multiple comparisons? Is pairwise comparison at each time interval the most efficient means of analysis in this design. Given these are essentially repeated measures, would repeated-measures ANOVA or a mixed model not be preferential.

To 2.) We thank the reviewer for his suggestion, to use a repeated-measures ANOVA for our data analysis. However, our target variables (i.e. EPC or MSC numbers) are not normally distributed, which would be one of the prerequisites for an ANOVA. Furthermore, the sizes of our subgroups are quite small. Because of these reasons it didn’t seem to be reasonable performing 

such a complex analysis as an ANOVA for repeated measurements.

 As this study has explorative character and because of the small sample sizes we didn’t correct for multiple comparisons (for the tests where 2 samples are compared). 

3.) Figure 1. I am a little confused as to the content of panel A. Is this at a specific time-point or is it some form of composite? Does it include day 0?

To 3.) Figure 1A comprises all time-points (including day 0). We first wanted to give an overview of the average cell numbers in the course of disease and then take a look at the different time-points in particular. If the reviewer feels that Figure 1A is too confusing to the reader and doesn’t add value to the understanding of the data, we are happy to take Figure 1A out.

4.) While highlighting the potential benefits of cell therapy in ARDS, in particular MSCs, the authors have failed to address important and emerging evidence of the harms of MSCs in ECMO, see:

Millar et al. Thorax 2019 10.1136/thoraxjnl-2017-211439

and

Cho et al. ASAIO J 2019 10.1097/MAT.000000000000102519

Evidence from complex large animal models which substantiate these claims have been presented in abstract form at a number of meetings.

To 4.) We thank the reviewer for his thoughtful comment and added the respective statement into the discussion section of our manuscript. It reads as follows:

Page 17, para 1:

 “However, the application of MSC in patients treated with ECMO needs further thorough clinical investigation. Recently, Millar et al. demonstrated a rapid decline in oxygenator performance in an ex-vivo model of ECMO [41]. Another group evaluated the viability and activity of MSC in ex vivo circulation conditions with and without an oxygenator (Cho et al.). The viability and activity of MSC decreased significantly when the oxygenator was used, mainly due to lysis and to the nonphysiologic condition itself (Cho et al.). Therefore, further studies need to evaluate MCS viability and activity in in-vivo models of ECMO.”

Minor critiques:

5.) On line 23 oxygenation index (OI) appears to have been confused with P/F ratio.

To 5.) Thank you for this indication. In German the P/F ratio is also refered to as oxygenation ratio, therefore we have confused the terminology here. We did not mean to refer to the oxygenation index and apologize for this inconvenience. The manuscript was adapted accordingly.

6.) This is of more minor importance given definitions are disputed, however the cell population characterised as MSCs in this paper are MSC-like when using the ISCT definition (10.1080/14653240600855905).

To 6.) We added a respective note into the methods section. It reads as follows.

 Page 7, para 1:

“Notably, definitions of MSC are disputed and the populations defined as MSC in our work might also be called “MSC-like” according to the definition of the International Society for Cellular Therapy [21].”

7.) The Lehle et al. study has been used incorrectly in varying parts of the manuscript to support the presence of circulating MSCs during ECMO. This study had no control group and in fact the study was based on cells adherent to the membrane oxygenator.

To 7.) We agree with the reviewer. We have now changed the wording in the corresponding parts, pointing out that the respective studies only suggest that ECMO might mobilize EPC and MSC. We hope, the reviewer agrees to the new statements, which read as follows:

Page 3, para 4:

“Furthermore, previous studies suggested that ECMO support might mobilize EPC and MSC into the circulation [11–13].

8.) Was this all VV ECMO? It would be beneficial to also have some idea of the differences in ventilatory settings between patients on or nor on ECMO. Perhaps this could be included as supplementary table.

To 8.) The patients included in our study were all supported with VV ECMO. Our study was intended to provide pilot data on ECMO and stem cell mobilization in ARDS patients. Therefore, we did not analyze the ECMO or ventilator settings and its impact on EPC and MSC mobilization. The number of patients in our study was simply too low as to calculate statistical associations between ECMO/ventilatory settings and stem cell numbers. Also, observations relative to ECMO in our study may not relate to all ECMO devices and strategies. We acknowledged this fact in the limitations section in our revised manuscript, which now reads as follows:

Page 16, para 1:

“Furthermore, we want to acknowledge that our observations relative to ECMO may not relate to all ECMO devices and strategies. Also, the impact of ventilator settings is not known yet.”

 However, in the revised manuscript we show now data relevant to the initiation of ECMO (Table 2.). We have analyzed for both patients groups (ECMO vs non-ECMO) the P/F ratio, the ventilator settings and the days after onset in form of hospital days, days of MV before ECMO (ECMO group) and days of MV before admission (non-ECMO group). The P/F ratio was significantly lower in the ECMO group compared to the non-ECMO group, as expected, since this was one of the main parameters to initiate ECMO. The ventilator settings and the hospital days before MV showed no significant difference.

9.) The mortality difference is marked between groups, where there any patients in the ARDS non-ECMO group who were refused ECMO?”

To 9.) No, none of the patients in the non-ECMO group was refused ECMO support.

---

## [Decision Letter · Decision Letter 1]

19 Dec 2019

Increased mobilization of mesenchymal stem cells in patients with acute respiratory distress syndrome undergoing extracorporeal membrane oxygenation

PONE-D-19-21924R1

Dear Dr. Patry,

We are pleased to inform you that your manuscript has been judged scientifically suitable for publication and will be formally accepted for publication once it complies with all outstanding technical requirements.

With kind regards,

You-Yang Zhao

Academic Editor

PLOS ONE

Additional Editor Comments (optional):

Reviewers' comments:

Reviewer's Responses to Questions

**Comments to the Author**

1. If the authors have adequately addressed your comments raised in a previous round of review and you feel that this manuscript is now acceptable for publication, you may indicate that here to bypass the “Comments to the Author” section, enter your conflict of interest statement in the “Confidential to Editor” section, and submit your "Accept" recommendation.

Reviewer #1: All comments have been addressed

Reviewer #2: All comments have been addressed

2. Is the manuscript technically sound, and do the data support the conclusions?

Reviewer #1: Yes

Reviewer #2: Yes

3. Has the statistical analysis been performed appropriately and rigorously? 

Reviewer #1: I Don't Know

Reviewer #2: Yes

4. Have the authors made all data underlying the findings in their manuscript fully available?

Reviewer #1: Yes

Reviewer #2: Yes

5. Is the manuscript presented in an intelligible fashion and written in standard English?

Reviewer #1: Yes

Reviewer #2: Yes

6. Review Comments to the Author

Reviewer #1: All comments have been addressed

All comments have been addressed

All comments have been addressed

All comments have been addressed

Reviewer #2: The authors have satisfactorily addressed my comments. These data will make a useful contribution to the literature.

7. PLOS authors have the option to publish the peer review history of their article (what does this mean?). If published, this will include your full peer review and any attached files.

Reviewer #1: Yes: Colin Evans

Reviewer #2: Yes: Jonathan E Millar

---

## [Editor Report · Acceptance letter]

8 Jan 2020

PONE-D-19-21924R1 

Increased mobilization of mesenchymal stem cells in patients with acute respiratory distress syndrome undergoing extracorporeal membrane oxygenation 

Dear Dr. Patry:

I am pleased to inform you that your manuscript has been deemed suitable for publication in PLOS ONE. Congratulations! Your manuscript is now with our production department. 

With kind regards,

on behalf of

Dr. You-Yang Zhao 

Academic Editor

PLOS ONE